# Associations between Tobacco Use, Surges, and Vaccination Status over Time in the COVID-19 Era

**DOI:** 10.3390/ijerph20021153

**Published:** 2023-01-09

**Authors:** Brandon W. Reed, Arthur L. Brody, Andre Y. Sanavi, Neal Doran

**Affiliations:** 1Mental Health Care Line, VA San Diego Healthcare System, San Diego, CA 92161, USA; 2Department of Psychiatry, University of California San Diego, La Jolla, CA 92093, USA

**Keywords:** tobacco use disorder, tobacco dependence, veteran, COVID-19 pandemic, vaccinations

## Abstract

Because COVID-19 is a respiratory and cardiovascular disease, understanding behaviors that impact cardiopulmonary health, such as tobacco use, is particularly important. While early studies suggested no change in prevalence of tobacco use as COVID-19 emerged, pandemic fatigue, shifting levels of COVID-19 transmission, and vaccine availability have all changed since the start of the pandemic. The current study examined whether time, COVID-19 surges, and/or vaccination status were associated with likelihood of daily and non-daily tobacco use over the first 24 months of the pandemic. Data were obtained from electronic health records of healthcare visits (*n* = 314,787) to four Southern California VA healthcare systems. Multinomial logistic regression analyses indicated that the likelihood of reporting both daily and non-daily tobacco use (versus non-use) increased over time. Daily and non-daily tobacco use were less common at visits that occurred during COVID-19 surges, as well as among veterans vaccinated against COVID-19. Our findings provide new insight into changes of tobacco use patterns and correlates across the first two years of this pandemic, and understanding these associations may facilitate understanding of health-related behaviors and inform clinical treatment of tobacco use disorder during the COVID-19 pandemic.

## 1. Introduction

COVID-19 has had a tremendous social impact, leading to more than 1 million deaths in the United States and 6.5 million internationally [1]. Multiple pandemic-related factors have varied over time, creating an opportunity to evaluate potential links with health-related behaviors. Because COVID-19 is a respiratory and cardiovascular disease, behaviors that impact cardiopulmonary health (e.g., tobacco use) may be of particular interest. There have been a number of previous reports evaluating changes in tobacco use during the pandemic, which have produced mixed findings. For example, some studies have found no COVID-19-related change in prevalence in cigarettes per day in the United States [2], Mexico [3], France [4], or China [5]. In contrast, other studies have suggested that tobacco use has increased, partially in response to stress [6,7], while still others have reported negative associations between perceived susceptibility to COVID-19 and cigarette smoking [8,9]. Importantly, many of these studies evaluated tobacco use during the initial months of the pandemic and the context has changed considerably, including changes in efforts to implement public health interventions, vaccine availability, and waxing and waning of transmission rates. There is a need for additional research to evaluate whether tobacco use and other health behaviors shift in association with these changes.

Evidence suggests that some behaviors intended to protect against COVID-19 infection (e.g., physical distancing) have declined over time [10], perhaps as a result of pandemic fatigue, or behavioral fatigue associated with ongoing adherence to COVID-19 restrictions [11]. However, little is known about changes in tobacco use patterns over the course of the pandemic. To date, general findings have suggested few changes in tobacco use over the relatively short history of the early COVID-19 pandemic. As time passes, potential effects of pandemic fatigue may become more apparent and can be measured by changes in health-related behaviors.

Our group recently evaluated tobacco use in a sample of veterans during the first year of the pandemic [12]. Significant findings included that reporting of non-daily and daily tobacco use was less prevalent during peak COVID transmission (when compared to pre-COVID and early COVID tobacco use reporting), although interactions within the model (e.g., time by sex and time by age) were less clear. These results suggest that surges of COVID-19 prevalence may have motivated reductions in tobacco use, perhaps to mitigate risk against the likelihood or severity of a potential infection. However, this study only examined use rates through February 2021. In the interim, there have been additional COVID-19 waves, most notably with the recent Omicron variant. It is unknown whether the association between tobacco use prevalence and periods of increased COVID-19 transmission has been stable over time.

Other studies have suggested that people who use tobacco may have reduced or stopped use in response to COVID-19 [13]. Vaccination status is a possible correlate of tobacco use during the COVID-19 pandemic; those who prioritized getting vaccinated might be less inclined to engage in other health-risking behaviors such as tobacco use, possibly in an effort to protect against contracting COVID-19 or mitigating symptoms. In recent investigations using electronic health records [14] and surveys [15,16], tobacco users were more likely than non-users to report vaccine hesitancy, with one survey finding double the odds of vaccine hesitancy in tobacco users [15]. The higher hesitancy and lower vaccination rate among tobacco users is present despite the fact that tobacco smoking is associated with higher perceived susceptibility to COVID-19 infection [17]. Examination of vaccination rates following COVID-19 surges, as happened with the Omicron variant, has recently become available [18]. However, continuing to examine associations between vaccination status and high-risk health behaviors—such as tobacco use among veterans—is critical to our understanding of paradigm shifts in the social response to the pandemic.

The current study aims to examine the associations of unique, COVID-19 related factors (i.e., time since the start of the pandemic, vaccination status, and whether COVID-19 prevalence was surging) with tobacco use reporting during the first 24 months of this pandemic. This pandemic period has been like no other in modern history, creating an opportune time to investigate tobacco use patterns and health behaviors. Based on studies cited above, we hypothesized that (1) as time progressed from the start of the pandemic, veterans would be more likely to report daily and non-daily tobacco use, (2) veterans would be less likely to report daily and non-daily tobacco use during COVID-19 surges, and (3) veterans who reported receiving at least one vaccination would be less likely to endorse daily and non-daily tobacco use.

## 2. Materials and Methods

### 2.1. Data Extraction and Variables Used

Data were obtained from electronic health records stored in the VA’s Corporate Data Warehouse and included records of visits that occurred between 18 March 2020 and 28 February 2022 at four VA healthcare system sites in Southern California (i.e., San Diego, Loma Linda, Long Beach, and Greater Los Angeles). The start date was based on California Governor Gavin Newsom’s public announcement to raise COVID-19 awareness in California [19]. The dataset included only visits at which an annual tobacco use screening was completed (approximately 85% of all recorded visits). During these visits, VA patients were asked “Do you smoke cigarettes or use tobacco every day, some days, or not at all?”. Tobacco use was categorized into daily, non-daily, or non-use categories. Time represented the number of months since the start of the pandemic (i.e., coded as the number of months between 18 March 2020 and the date of the individual visit).

To capture the effect of COVID-19 surges, we created a binary variable reflecting whether or not the visit occurred during a surge in which COVID-19 infection rates were heightened. The first surge began on 16 November 2020 (i.e., when California Governor Newsom announced more stringent restrictions as case counts rose to nearly 10,000 per day [20]) and ended on 28 February 2021, corresponding to Governor Newsom reaching a school reopening compromise [21] as infection rates dropped. The second surge began 13 December 2021 (i.e., when Governor Newsom reinstituted the mask mandate for Californians [22]), and lasted until the end of the study on 28 February 2022. See Table 1 for an overview of coding of COVID-19 surges.

Vaccine data included documented, VA-provided COVID-19 vaccinations as well as self-reported vaccination that were received at non-VA locations. Because the maximum recommended number of COVID-19 vaccinations changed over the course of the study (i.e., the addition of boosters and differential recommendations for boosters as a function of age or health status), we coded whether or not veterans had received one or more vaccinations prior to the visit, resulting in a dichotomized categorization of veterans reporting being either vaccinated or non-vaccinated.

### 2.2. Data Analytic Plan

We used a multinomial logistic regression model to assess associations between tobacco use and time (in months) that had elapsed since the start of the pandemic, COVID-19 prevalence rate surges, and vaccination status. Multinomial logistic regressions are useful when analyzing categorical predictor variables (i.e., surges, vaccination status, etc.). Demographic and utilization variables (e.g., age, total number of visits during the study period, sex, race, ethnicity, and VA station location) were included as covariates. For the outcome variable, tobacco non-use was the reference category. All analyses were conducted using SPSS 28 software with α = 0.05.

## 3. Results

### 3.1. Descriptive Statistics

Data from electronic health records captured unique visits (*n* = 314,787) of veterans (*n* = 235,794) across four VA Hospitals in Southern California. The mean age was 57.9 (*SD* = 18.0) and nearly 9 in 10 were male (89.7%). Nearly 70% of veterans reported being non-Hispanic. Over half of veterans were Caucasian (52.7%), 16.7% identified as Black or African American, 6.9% were Asian American, 2.2% were Native Hawaiian or Pacific Islander (NH/PI), 1.3% were American Indian or Alaskan Native (AI/AN), and 15.6% identified as “other” or declined to provide race data. Regarding tobacco use patterns, 10.8% of veterans reported daily use and 4.2% reported non-daily use. Nearly 40% of visits were with veterans who had reported receiving one or more COVID-19 vaccine doses prior to the visit, and 25.7% of all visits occurred during a COVID-19 surge. To address the possibility that vaccine availability would affect the association between vaccine status and tobacco use, the multinomial logistic regression was reanalyzed and included only visits (*n* = 210,640) in which the vaccine was locally available (i.e., on or after 1 December 2021). The pattern of results of the reanalysis remained unchanged (see Appendix A).

See Table 2 for tobacco use status by group. There were significant between group differences found between: males versus females (*χ*^2^ = 425.56, *p* < 0.001); Hispanic vs. non-Hispanic (*χ*^2^ = 991.13, *p* < 0.001); race (*χ*^2^ = 1486.69, *p* < 0.001); COVID-19 surge period vs. non-surge period (*χ*^2^ = 19.18, *p* < 0.001); vaccinated vs. non-vaccinated (*χ*^2^ = 585.58, *p* < 0.001); VA location (*χ*^2^ = 1285.44, *p* < 0.001); age (*F* = 1660.76, *p* < 0.001); time since the start of the pandemic (*F* = 8.48, *p* < 0.001); and visit number (*F* = 32.49, *p* < 0.001). Multinomial model results are shown in Table 3, and the overall model was significant (*χ*^2^ = 6319.25, *df* = 26, *p* < 0.001).

### 3.2. Associations with Daily Tobacco Use

In comparing daily tobacco use to the reference group of tobacco non-use, time since the pandemic began was a significant predictor of daily tobacco use. Over the 24-month study period, veterans were 1.5% more likely to report daily tobacco use with each passing month. In contrast, there was an inverse association between daily tobacco use and surge, such that visits that occurred during COVID-19 surges were associated with 11% lower likelihood of daily tobacco use compared to visits that had not occurred during a surge. Being vaccinated was associated with a 30% lower likelihood of reporting daily tobacco use compared to those who did not report having been vaccinated.

There were also associations between daily tobacco use and several demographic variables, indicating that daily tobacco use was more likely among those who identified as Black, male, and/or non-Hispanic, and among younger veterans. Veterans who identified as Asian were less likely to be daily users.

### 3.3. Associations with Non-Daily Tobacco Use

Similar to daily use reporting, there was a significant positive association between time that had elapsed from the start of the pandemic and non-daily tobacco use reporting, such that each passing month was associated with 0.8% higher likelihood of non-daily tobacco use. Whether or not the visit occurred during a COVID-19 surge was also inversely associated with non-daily tobacco use; veterans were 7% less likely to report non-daily use during COVID-19 surges. Veterans who had received any vaccine were 14% less likely to endorse non-daily tobacco use.

Non-daily tobacco use was more commonly endorsed by veterans who identified as Black and male, and by younger veterans. Asian veterans were less likely than White veterans to report non-daily use. Refer to Table 3 for a summary of the results.

## 4. Discussion

The primary aims of this study were to examine whether time, COVID-19 surges, and vaccination status were associated with self-reported tobacco use during the COVID-19 pandemic. Consistent with our primary hypothesis, time since the pandemic began was significantly positively associated with both daily and non-daily tobacco use. We speculated that initial reporting of the novel coronavirus—and the associated pulmonary issues and complications—may have motivated some initial reductions in use [12]. However, pandemic fatigue, or fatigue associated with the persistence of the coronavirus prevalence, likely increased with time and may partially explain increased daily and non-daily tobacco reporting as months progressed. This finding is consistent with previous evidence indicating that adherence to COVID-19 health precautions, such as physical distancing and staying at home, declined as the pandemic progressed [10].

While tobacco use reporting increased with time since the start of the pandemic, reporting of both daily and non-daily tobacco use was lower during periods of COVID-19 surges. This finding is consistent with our previous work [12], suggesting that individuals who reported tobacco use may be more amenable to short term changes and may be more compelled to adjust heath behavior in acute, high-risk situations, but may then return to baseline tobacco use as pandemic fatigue increases. A recent survey found that tobacco smoking was associated with higher perceived susceptibility to COVID-19 infection [17]. This perception could be explained in part by worsening fear and increased health consciousness as media reports of COVID-19-related deaths and hospitalizations increased during COVID-19 surges with increased prevalence of infection rates. Additionally, these surges necessitated intensified lock down procedures (i.e., stay-at-home mandates, six-foot distancing requirements) to reduce person-to-person contact [23] and changed the landscape of social interactions. Some veterans may have reported less tobacco use during surges due to greater restrictions on their ability to interact socially with users, or limits impacting ease of obtaining tobacco.

Veterans who received at least one dose of the coronavirus vaccine were 20% and 11% less likely to report daily and non-daily use, respectively. Previous literature suggests that, relative to never and former smokers, current smokers reported significantly greater hesitancy and concerns, and were more likely to be unwilling or uncertain about receiving a COVID-19 vaccine [16]. Health consciousness could represent a reasonable explanation for this effect, as it is possible that veterans who are more likely to seek the coronavirus vaccine may also be more likely to abstain from tobacco use to avoid negative consequences. In a study focusing on tobacco users, approximately 40% were unvaccinated and those who were vaccine hesitant expressed concerns about the lack of research on the vaccine, distrust of the safety of the vaccine, and fears about side effects [19]. As the coronavirus is in part a respiratory disease, those who took preventative measures against infection severity (i.e., getting vaccinated) may also be less likely to engage in other potentially unhealthy behaviors, such as tobacco use.

Results of our study can be used to better inform the healthcare management of our veteran population. VA providers may now be better able to evaluate tobacco use among their patients and provide information regarding associations between tobacco use and COVID-19-era factors, such as vaccination status and surges. Discussing risks of increased tobacco use (and tendencies toward decreasing healthy behaviors) over time will help prevent the effects of pandemic fatigue and may lead to sustained efforts toward tobacco reduction or abstinence, particularly as prevalence rates continue to ebb and flow with new variant surges. Policy changes to increase patient education can serve as prevention efforts to help mitigate risks associated with continued tobacco use.

There were several limitations to our study that may impact the generalizability of the findings. Data were drawn from electronic health records of veterans receiving VA care in the southwest region of the United States. Data are subject to the inherent limitations of all research capitalizing on electronic medical records, namely that data were originally collected for non-research purposes and may suffer from diagnostic inaccuracy and may fail to account for all confounding factors [24]. Assessment of tobacco use, while routine, is relatively brief and the reliability and validity of our results are partially contingent on the degree to which providers were judicious in accurately collecting and reporting these data in the electronic health record. Relatedly, we were unable to explore more detailed inquiries related to both tobacco use or its predictors, such as use of specific products; motivations for tobacco use or cessation; and motivations for vaccine hesitancy or compliance. In addition, while southwestern VA hospitals serve a diverse range of cultural, racial, ethnical, and socioeconomic populations, the demographics of patients who provided our data may not be fully representative of the general population. While access to healthcare was restricted during the pandemic, particularly during the initial months of COVID-19 emergence [25], VA hospitals remained open with both in-person and telehealth visits made available to the veterans they serve. Data were retrieved from records of veterans who attended a healthcare visit, which may not fully represent the general population, including those who did not seek or attend health visits during the pandemic period.

## 5. Conclusions

This study explored associations between the time since the start of the COVID-19 pandemic, surges, and vaccination status on daily and non-daily tobacco use. Consistent with our hypotheses, we found a significant positive association between time and tobacco use reporting, that veterans would report less tobacco use during COVID-19 surges, and that veterans who reported being vaccinated would also report less tobacco use. Future studies should explore within group differences on tobacco use status (i.e., within-person changes over time) and could include a more detailed assessment of tobacco use patterns, motives, and changes as well as further investigate the role of vaccination seeking or hesitancy on tobacco use outcomes. Additionally, future studies should evaluate factors pertaining to other substance use and mental health concerns, with a focus on subpopulation disparities and vulnerabilities, as these may have changed during the course of the pandemic. These studies of tobacco use and related factors would continue to help providers educate patients of the risks of tobacco use during the pandemic era and inform leadership to enact policy changes help to optimize the practice of treating and preventing nicotine dependence for as long as COVID-19 continues to persist.

## Figures and Tables

**Table 1 ijerph-20-01153-t001:** Major events representing COVID-19 prevalence rate changes in California.

Time Period	Associated California Initiating Event
18 March 2020–15 November 2020	COVID-19 awareness announced
16 November 2020–28 February 2021 *	Increased restrictions as daily cases near 10,000
1 March 2021–12 December 2021	Relative latency in COVID-19 transmission
13 December 2021–28 February 2022 *	Return of mask mandate, Omicron variant emergence

* Coded as a COVID-19 surge.

**Table 2 ijerph-20-01153-t002:** Group demographics by tobacco use status.

	No Use	Daily Use	Non-Daily Use
Visits (*n*)	268,041	33,770	12,976
Race [%]			
White/Caucasian	85.4%	10.9%	3.6%
Black/AA	80.7%	13.5%	5.7%
Asian American	88.6%	7.9%	3.4%
NH/PI	84.9%	10.5%	4.7%
AI/AN	84.8%	10.5%	4.6%
Other	87.4%	8.3%	4.3%
Ethnicity [%]			
Hispanic	87.7%	7.4%	4.9%
Non-Hispanic	84.4%	11.8%	3.8%
Sex [%]			
Male	84.7%	11.0%	4.3%
Female	89.0%	8.2%	2.7%
Age [mean (*SD*)]	58.5 (18.22)	55.2 (15.50)	50.49 (16.50)
Time [mean (*SD*)]	11.58 (6.49)	11.44 (6.49)	11.45 (6.49)
Vaccination status [%]			
Vaccinated	87.1%	9.4%	3.6%
Non-vaccinated	83.9%	11.6%	4.5%
Surge [%]			
COVID Surge	85.6%	10.4%	4.0%
No COVID Surge	85.0%	10.8%	4.2%
Visit Number [mean (*SD*)]	1.26 (0.46)	1.25 (0.47)	1.23 (0.46)
VA Location			
VA Long Beach	86.9%	8.8%	4.2%
VA Loma Linda	86.7%	10.3%	3.0%
VA San Diego	87.2%	8.8%	4.0%
VA Greater Los Angeles	82.3%	13.1%	4.6%

Note: AA = African American, NH = Native Hawaiian, PI = Pacific Islander, AI = American Indian, AN = Alaskan Native.

**Table 3 ijerph-20-01153-t003:** Multinomial model predicting daily and non-daily tobacco use.

Predictor	Daily Tobacco Use vs. Nonuse				Non-Daily Tobacco Use vs. Nonuse			
	Coeff.	SE	O.R.	95% CI	Coeff.	SE	O.R.	95% CI
Intercept	−2.94	0.12	-	-	−3.57	0.19	-	-
Time	0.02	<0.01	1.02	1.01, 1.02	0.01	<0.01	1.01	1.00, 1.01
Vaccination status	−0.30	0.02	0.75	0.72, 0.77	−0.14	0.03	0.87	0.82, 0.92
Surge	−0.11	0.02	0.90	0.87, 0.93	−0.08	0.03	0.93	0.82, 0.97
Race								
Black/AA	0.17	0.02	1.18	1.14, 1.22	0.50	0.03	1.66	1.57, 1.74
Asian American	−0.53	0.03	0.59	0.55, 0.62	−0.22	0.04	0.81	0.74, 0.88
NH/PI	−0.10	0.05	0.91	0.83, 0.99	0.15	0.07	1.17	1.02, 1.33
AI/AN	0.01	0.06	1.01	0.91, 1.13	0.09	0.09	1.10	0.92, 1.30
Other	−0.21	0.03	0.81	0.77, 0.85	0.02	0.04	1.02	0.95, 1.09
Ethnicity								
Hispanic	−0.58	0.02	0.56	0.54, 0.58	0.05	0.03	1.06	1.00, 1.11
Sex	−0.53	0.03	0.59	0.56, 0.62	−0.90	0.04	0.41	0.37, 0.44
Age	−0.02	<0.01	0.99	0.98, 0.99	−0.03	<0.01	0.97	0.97, 0.97
Visit Number	0.02	0.02	1.02	0.99, 1.06	−0.08	0.03	0.93	0.88, 0.98
VA location	<0.01	<0.01	1.00	1.00, 1.00	<0.01	<0.01	1.00	1.00, 1.00

Note: The following categories served as reference groups: no history of vaccination, visit not occurring during a COVID−19 surge, being Caucasian, being non-Hispanic, and being male.

## Data Availability

The data are not publicly available. A deidentified dataset can be obtained by contacting the authors.

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
