# Peer review of "Associations between Tobacco Use, Surges, and Vaccination Status over Time in the COVID-19 Era"

_ijerph, 2023, doi:10.3390/ijerph20021153_

Round 1

Reviewer 1 Report (Previous Reviewer 1)

The authors have responded to the observations made, substantially improving the article and it can be be considered suitable for publication in the journal

Author Response

The authors have responded to the observations made, substantially improving the article and it can be considered suitable for publication in the journal.
Thank you for this comment and taking the time to help with the review of this article.

Reviewer 2 Report (Previous Reviewer 2)

I would like to thank the authors for the thoroughly examination of the comments and addressing each of them.

A few minor comments that would improve this manuscript can be found below.

1)     Line 78: add the population of study “….with veterans tobacco use…”

2)     Line 135: add the second analysis as supplementary material

3)     Line 156 - 157, 159, 160, 169, 171 and 172. Please, specify to what are you comparing to (i.e. "higher likelihood of stopping tobacco use in vaccinated, compared to unvaccinated participants...")

Author Response

I would like to thank the authors for the thoroughly examination of the comments and addressing each of them.
A few minor comments that would improve this manuscript can be found below.
1) Line 78: add the population of study “….with veterans tobacco use…”
We have added the population of the current study to line 74.
2) Line 135: add the second analysis as supplementary material
The analysis has been added as supplementary material.
3) Line 156 - 157, 159, 160, 169, 171 and 172. Please, specify to what are you comparing to (i.e. "higher likelihood of stopping tobacco use in vaccinated, compared to unvaccinated participants...")
This section has been revised to clarify the comparison groups.

Reviewer 3 Report (Previous Reviewer 3)

Thank you for the opportunity to review this manuscript. The authors did a great job addressing the reviewer’s comments. I have no additional comments. 

Author Response

Thank you for the opportunity to review this manuscript. The authors did a great job addressing the reviewer’s comments. I have no additional comments.

Thank you for this comment and taking the time to help with the review of this article.

Reviewer 4 Report (New Reviewer)

1. Change the paper title to - "Tobacco Use Trends Among Veterans During the COVID-19 Pandemic"

2. Discuss the major implications for practice, prevention, policy, and public health.

3. Provide directions for future research.

4. Elaborate on limitations (e.g. as it relates to validity and reliability of findings).

5. Discuss the role of various factors (e.g. mental health and tobacco use or vaccinated ppl being more cautious or proactive, etc).

6. Tables should be self-explanatory and include actual number of study participants (e.g. Table 1 is missing a lot of this information). Also, comment on statistical model fits and why they were selected?

Author Response

1. Change the paper title to - "Tobacco Use Trends Among Veterans During the COVID-19 Pandemic"
I‘d like to thank the author for the suggestion. Based on another reviewer’s feedback, we have decided to avoid using the word ‘trends’ because we are not discussing change at the individual level. However, the title has been changed to clearly reflect the period of the study having occurred in the COVID-19 ERA.
2. Discuss the major implications for practice, prevention, policy, and public health.
We added a paragraph (line 220) to address these implications.
3. Provide directions for future research.
Additional directions for future research can be found on lines 258-264
4. Elaborate on limitations (e.g. as it relates to validity and reliability of findings).
We elaborated our study limitations on lines 234-237
5. Discuss the role of various factors (e.g. mental health and tobacco use or vaccinated ppl being more cautious or proactive, etc).
We’ve expanded the discussion section to further elaborate on these factors and how their implications on our study, as well as future studies.
6. Tables should be self-explanatory and include actual number of study participants (e.g. Table 1 is missing a lot of this information). Also, comment on statistical model fits and why they were selected?
The title of table 1 has been change to better inform readers of the purpose of these time frames and coding them as surges or non-surges of COVID rates. Table 2 has included the number of study participants. We commented on model fit on line 153, and provided explanation of why it was selected on line 121.

Round 2

Reviewer 4 Report (New Reviewer)

Thanks for the revisions 

Author Response

Thank you for this comment and taking the time to help with the review of this article.

This manuscript is a resubmission of an earlier submission. The following is a list of the peer review reports and author responses from that submission.

Round 1

Reviewer 1 Report

Although the article deals a topic of great relevance, and this is implied in the review of the bibliography, it lacks interest as it is proposed. Although it relates the use of tobacco during the pandemic and the possibilities for users to report it, what is the purpose of it? Likewise, what was the tobacco consumption of users before the pandemic? Without this, the results cannot be compared and it is not possible to draw firm conclusions about the greater or lesser possibility of informatios on consumption. On the other hand, what is the interest of the origin of the participants if there are no differences are not established between them,  as well as according to age and sex? It is considered necessary to complete the study in order to be published in this journal.

Author Response

Reviewer 1:

Although the article deals a topic of great relevance, and this is implied in the review of the bibliography, it lacks interest as it is proposed. Although it relates the use of tobacco during the pandemic and the possibilities for users to report it, what is the purpose of it? Likewise, what was the tobacco consumption of users before the pandemic? Without this, the results cannot be compared and it is not possible to draw firm conclusions about the greater or lesser possibility of information on consumption. On the other hand, what is the interest of the origin of the participants if there are no differences are not established between them, as well as according to age and sex? It is considered necessary to complete the study in order to be published in this journal.

We would like to thank the reviewer for these comments. In the introduction, we have clarified the goals of the paper to emphasize that we are examining tobacco use patterns, and changes in tobacco use prevalence, over the course of the pandemic and not focusing on individual change pre- and post-pandemic (lines 76-80). We have included a new table that illustrates between group differences (line 142) as well as reported group comparisons at the end of section 3.1 (line 147).

Reviewer 2 Report

First, I want to thank the authors for their manuscript named “Relationships between Tobacco Use in the COVID-19 Era and Time, Surges, and Vaccination Status”. This draft examined the association between time, COVID-19 surges, and vaccination with the likelihood of daily and non-daily tobacco use over the 24 months of the pandemic in veterans.

Below you can find some of the comments and questions regarding this manuscript:

1.     In a few parts of the manuscript as well as in the title, the authors used the term “relationship”, considering the study design, the appropriate term is association instead of relationship, as a relationship implies causality, and this is a cross-sectional study.

2.     A descriptive table of the sample use is needed, please stratified the descriptive characteristics between non, daily and non-daily tobacco use, and apply the necessary statistical tests to check for differences among the groups.

3.     I would suggest accounting for other substances in the model (i.e., alcohol, cannabis, and opioid use) and well as potential mental health concerns (depression, PTSD, anxiety), these factors can be linked to both tobacco use and strenuous periods such as the pandemic

4.     I am wondering if there was a change in the number of visits in the surge compared to the regular periods. Is there any difference in the proportion of tobacco use between surge vs non-surge periods?

5.     Line 160-163: “Consistent with our primary hypothesis, time since the pandemic began was predictive of both daily and non-daily tobacco use. We speculate that initial reporting of the novel coronavirus – and the associated pulmonary issues and complications – may have motivated some initial reductions in use”. This statement suggest that tobacco use should have decreased since the start of COVID-19, but the results showed marginal higher odds of tobacco use reporting with time increasing. This is an assumption, please use a reference to support this statement. 

6.     Line 169-170: “While tobacco use trended up with time, reporting of both daily and non-daily tobacco use was lower during periods of COVID-19 surges.”  Please, be careful with the phrasing of this statement. Time trending was not really measured with the odds ratios that were calculated, as there is no information of tobacco use by month.

7.     Line 219-220: “Findings from the current study and future studies may reveal sub-population disparities and vulnerabilities, which will inform future treatment for tobacco use.” In what capacity the results obtained in a strenuous period can be applied to treatment at regular basis?

Author Response

Reviewer 2:

First, I want to thank the authors for their manuscript named “Relationships between Tobacco Use in the COVID-19 Era and Time, Surges, and Vaccination Status”. This draft examined the association between time, COVID-19 surges, and vaccination with the likelihood of daily and non-daily tobacco use over the 24 months of the pandemic in veterans.

We would like to give thanks to Reviewer 2 for the time spent on reviewing this article, as well as having provided meaningful comments that we believe strengthen the impact of this manuscript.

Below you can find some of the comments and questions regarding this manuscript: 

  1. In a few parts of the manuscript as well as in the title, the authors used the term “relationship”, considering the study design, the appropriate term is association instead of relationship, as a relationship implies causality, and this is a cross-sectional study.

We have replaced “relationship” with “association” throughout the manuscript to prevent confusion that we are describing causality, including in the title, in the abstract, and in the discussion. For examples, please see lines 2, 21, and 73. We also removed other language that suggested implications of causality throughout the paper.

  1. A descriptive table of the sample use is needed, please stratified the descriptive characteristics between non, daily and non-daily tobacco use, and apply the necessary statistical tests to check for differences among the groups.

Thank you for this suggestion. We have included a demographics table (line 142, stratified by tobacco use category, to clarify the nature of between group differences. The text now contains a report of statistical tests to check for differences between groups (line 147).

  1.  

While we agree that these factors would be interesting to include in our modeling, our focus was on tobacco use and COVID surges, vaccination status, and time since the start of the pandemic. While we do not have the data to explore associations between other predictors and tobacco use, we agree that this would be interesting to examine in future studies. Language was included in the future studies section of the conclusion to reflect this interest (line 243).

  1. I am wondering if there was a change in the number of visits in the surge compared to the regular periods. Is there any difference in the proportion of tobacco use between surge vs non-surge periods?

We hope that the inclusion of the demographics table and the statistical tests of group differences have addressed this comment (lines 142, 147, respectively).

5.

On line 191, we clarify reasoning that people who use tobacco may be inclined to make short term changes in response to acutely distressing situations, such as during COVID-19 surges, but may then return to baseline as the pandemic progresses.

  1. Line 169-170: “While tobacco use trended up with time, reporting of both daily and non-daily tobacco use was lower during periods of COVID-19 surges.”  Please, be careful with the phrasing of this statement. Time trending was not really measured with the odds ratios that were calculated, as there is no information of tobacco use by month.

Thank you for this feedback. We clarified that our time variable was measured in months since the start of the pandemic on lines 96 and 119. We removed “trending” when discussing time and instead clarified that tobacco reporting was higher during COVID-19 surges.

  1. Line 219-220: “Findings from the current study and future studies may reveal sub-population disparities and vulnerabilities, which will inform future treatment for tobacco use.” In what capacity the results obtained in a strenuous period can be applied to treatment at regular basis?

At the end of the conclusions section (line 247), we added that the results of our study can be applied to treatment as long as the COVID-19 pandemic continues to persist and remains a factor in tobacco use reporting.

Reviewer 3 Report

According to the authors, the aim of this study was to examine whether time, COVID-19 surges, and/or vaccination status were associated with likelihood of daily and non-daily tobacco use over the first 24 months 14 of the pandemic.

Even though several publications report smoking behaviors during the pandemic, this paper addresses tobacco use among a specific population (sample of Veterans) and for a longer period of time. It is well-written and certainly of high interest and importance to researchers and clinicians. Some notes:

1)      It is still interesting that tobacco use increased over time but not during the surge time. The dataset included only visits at which an annual tobacco use screening was completed. Is it possible that providers were not completing this assessment in visits that occurred during COVID-19 surges to shorten the contact with patients? Do you know how many patients were missing the tobacco assessment? Did it differ across time points? How does this result compare to other tobacco studies during the COVID-19 surges?

2)      Since the vaccine was not available to everyone at the begging of the pandemic, it is possible that the vaccine was not available before the visit. How many patients had the visit between 3/18/20 to 2/28/2021? How did you address these cases?

3)      The results of this paper showed that VA smokers increased their tobacco use over the pandemic but not during the period of surges. One hypothesis is that they were concerned about complications of COVID due to smoking. However, they were less likely to take the vaccine, which also could prevent from worsening the COVID infection. The authors discussed both results, but it still sounds like conflicting results to me.

Author Response

Reviewer 3:

According to the authors, the aim of this study was to examine whether time, COVID-19 surges, and/or vaccination status were associated with likelihood of daily and non-daily tobacco use over the first 24 months 14 of the pandemic.

Even though several publications report smoking behaviors during the pandemic, this paper addresses tobacco use among a specific population (sample of Veterans) and for a longer period of time. It is well-written and certainly of high interest and importance to researchers and clinicians. Some notes:

1) It is still interesting that tobacco use increased over time but not during the surge time. The dataset included only visits at which an annual tobacco use screening was completed. Is it possible that providers were not completing this assessment in visits that occurred during COVID-19 surges to shorten the contact with patients? Do you know how many patients were missing the tobacco assessment? Did it differ across time points? How does this result compare to other tobacco studies during the COVID-19 surges?

We clarified that all visits included in the study included a tobacco assessment. (see line 93), and that the completion rate for these tobacco assessments were approximately 85% Also, please see newly included Table 2 (line 142) for demographics stratified by tobacco use, as well as a report of group differences on line 147.

2) Since the vaccine was not available to everyone at the beginning of the pandemic, it is possible that the vaccine was not available before the visit. How many patients had the visit between 3/18/20 to 2/28/2021? How did you address these cases?

Thank you for this comment. Earliest vaccine availability date was in December of 2021. The model was reanalyzed excluding visits that occurred before this time, and the pattern of results remained unchanged. We included a statement regarding this concern on line 135.  

3) The results of this paper showed that VA smokers increased their tobacco use over the pandemic but not during the period of surges. One hypothesis is that they were concerned about complications of COVID due to smoking. However, they were less likely to take the vaccine, which also could prevent from worsening the COVID infection. The authors discussed both results, but it still sounds like conflicting results to me.

We clarified that users of tobacco may be more amenable to short term changes and may be more compelled to adjust behavior in acute, high-risk situations, with an added citation (see lines 191-194).